# *Euglena Gracilis* and β-Glucan Paramylon Induce Ca^2+^ Signaling in Intestinal Tract Epithelial, Immune, and Neural Cells

**DOI:** 10.3390/nu12082293

**Published:** 2020-07-30

**Authors:** Kosuke Yasuda, Ayaka Nakashima, Ako Murata, Kengo Suzuki, Takahiro Adachi

**Affiliations:** 1Euglena Co., Ltd., Tokyo 108-0014, Japan; kosuke.yasuda@euglena.jp (K.Y.); ako.murata@euglena.jp (A.M.); suzuki@euglena.jp (K.S.); 2Department of Immunology, Medical Research Institute, Tokyo Medical and Dental University, Tokyo 113-8510, Japan

**Keywords:** β-1,3-glucan, *Euglena gracilis*, Ca^2+^ signaling, intestinal epithelial cell, intravital imaging, small intestine, immune system

## Abstract

The intestinal tract contains over half of all immune cells and peripheral nerves and manages the beneficial interactions between food compounds and the host. Paramylon is a β-1,3-glucan storage polysaccharide from *Euglena gracilis* (*Euglena*) that exerts immunostimulatory activities by affecting cytokine production. This study investigated the signaling mechanisms that regulate the beneficial interactions between food compounds and the intestinal tract using cell type-specific calcium (Ca^2+^) imaging in vivo and in vitro. We successfully visualized *Euglena*- and paramylon-mediated Ca^2+^ signaling in vivo in intestinal epithelial cells from mice ubiquitously expressing the Yellow Cameleon 3.60 (YC3.60) Ca^2+^ biosensor. Moreover, in vivo Ca^2+^ imaging demonstrated that the intraperitoneal injection of both *Euglena* and paramylon stimulated dendritic cells (DCs) in Peyer’s patches, indicating that paramylon is an active component of *Euglena* that affects the immune system. In addition, in vitro Ca^2+^ imaging in dorsal root ganglia indicated that *Euglena*, but not paramylon, triggers Ca^2+^ signaling in the sensory nervous system innervating the intestine. Thus, this study is the first to successfully visualize the direct effect of β-1,3-glucan on DCs in vivo and will help elucidate the mechanisms via which *Euglena* and paramylon exert various effects in the intestinal tract.

## 1. Introduction

The intestinal tract is the first line of defense against pathogenic microorganisms and manages beneficial interactions between food compounds and the host [1]. These interactions are mediated by the neural, endocrine, and immune systems to maintain intestinal homeostasis [2]; however, homeostatic dysfunction can significantly affect host immunity and the course of chronic inflammation [3]. Some probiotics and polysaccharides have been found to regulate intestinal and immune homeostasis; for instance, probiotic *Bifidobacterium bifidum* alleviates dysbiosis and constipation in mice induced by a low-fiber diet [4], while grains fermented with *Aspergillus oryzae* can protect against chronic constipation and gastrointestinal damage [5,6]. In addition, the soluble dietary fiber β-1,3-glucan from seaweed has been shown to suppress intestinal inflammation in mouse models of human inflammatory bowel disease by increasing the *Lactobacillus* population and the number of regulatory T cells in the colon [7]. However, different polysaccharides can exert varying effects on immune homeostasis [8,9,10].

*Euglena gracilis* (*Euglena*) is a microalga that contains a wide range of nutrients, including vitamins, minerals, amino acids, and fatty acids. Since it combines the properties of both plants and animals, it is often used as a food or dietary supplement [11]. The storage polysaccharide paramylon is an insoluble dietary fiber unique to *Euglena* that has a triple-helical polymer structure composed of straight-chain β-1,3-glucans. Like other β-1,3-glucans, paramylon has various beneficial effects on health, such as modulating immune function [12,13,14,15] and suppressing visceral fat accumulation, likely by improving the intestinal environment [16,17,18,19]. In addition, paramylon can bind to and stimulate Dectin-1, the primary receptor on epithelial cells, macrophages, and dendritic cells (DCs) to exert immunomodulatory effects [20,21,22]. Following their ingestion, polysaccharides and probiotics are thought to be taken up into M cells, a type of intestinal epithelial cell (IEC) that lies over Peyer’s patches (PPs), allowing them to access gut immune cells such as macrophages and DCs [23,24]; however, it is difficult to monitor these biological events in real time.

Calcium imaging using genetically encoded calcium biosensors has enabled nutrition-sensing mechanisms in the intestinal tract to be visualized. For example, Calcium ion (Ca^2+^) signaling in specific cell populations has been visualized in conditional transgenic mice expressing Yellow Cameleon 3.60 (YC3.60) [25,26]. Ca^2+^ is a universal secondary messenger that performs multiple functions in most cells, including lymphocytes, epithelial cells, and neurons [27,28,29,30]. Intravital Ca^2+^ imaging in transgenic mice expressing YC3.60 under the control of the CD11c gene promoter has demonstrated that oral propolis administration stimulates DCs in lymphoid tissues [31], while in vivo Ca^2+^ imaging has revealed that probiotics induce Ca^2+^ signaling in IECs under physiological conditions [32]. Enteroendocrine cells are a form of IEC that constitute the largest endocrine organ in the human body, while the autonomic nervous system that innervates the intestine exerts strong modulatory effects on the motor, secretory, and immunologic functions of the intestinal tract [33,34]. To elucidate the complex sensing mechanisms that recognize *Euglena* and paramylon in the intestinal tract, we performed cell type-specific Ca^2+^ imaging in three conditional transgenic mouse lines with specific or ubiquitous YC3.60 expression [26,31] to visualize *Euglena*- and paramylon-induced Ca^2+^ signaling in IECs, DCs in intestinal PPs, and nerve cells innervating the intestine.

## 2. Materials and Methods

### 2.1. Animals

Conditional YC3.60 expressing transgenic mice were as described previously [26]. The floxed YC3.60 reporter (YC3.60^flox^) mouse line was crossed with cell type-specific Cre mouse lines (CD11c-Cre, Nestin-Cre, and CAG-Cre) to produce YC3.60^flox^/CD11c-Cre, YC3.60^flox^/Nestin-Cre, and YC3.60^flox^/CAG-Cre mice, respectively [26,31]. These mice were maintained in our animal facility under specific pathogen-free conditions in accordance with the animal care guidelines of the Tokyo Medical and Dental University, while animal procedures were approved by its Animal Care Committee (approval number A2019-207C4, date of approval 3 December 2019).

### 2.2. Dorsal Root Ganglia (DRG) Cells

Dorsal root ganglia (DRG) cells were prepared from YC3.60^flox^/Nestin-Cre mice as described previously [35]. Briefly, mice were euthanized by cervical dislocation and their DRG excised before being treated with collagenase (1 mg/mL) and trypsin (0.25 mg/mL), as described previously [36]. The cells were then washed twice with Dulbecco’s modified Eagle’s medium (DMEM) and cultured at 37 °C on a gelatin-coated plate with DMEM containing 10% fetal calf serum and 100 ng/mL of nerve growth factor.

### 2.3. Test Substances

*Euglena* powder and paramylon were obtained from Euglena Co., Ltd. (Tokyo, Japan). The nutritional composition of the *Euglena* powder was as follows: carbohydrates 55.0%, protein 29.9%, and lipid 9.0%. Approximately 70–80% of the carbohydrate content was paramylon. Paramylon was prepared according to the standard method, as follows: cultured *Euglena gracilis Z* cells were collected by continuous centrifugation and washed with water. After being suspending in water, the cells were disintegrated using ultrasonic waves and the cell contents (containing paramylon) were collected. The crude paramylon preparation was treated with 1% sodium dodecyl sulfate (SDS) solution for 1 h at 95 °C followed by 0.1% SDS for 30 min at 50 °C to remove lipids and proteins. After centrifugation, paramylon was refined by repeated washing with water, acetone, and ether.

### 2.4. Intravital and In Vitro Imaging

IECs and PPs from anesthetized mice were imaged as described previously [32]. Small intestinal tracts were surgically opened lengthwise, placed on a glass cover slip, and immobilized on a microscope stage. To observe IECs, 0.1 mL of *Euglena* or paramylon in phosphate buffered saline (PBS; 1 mg/mL) was added to the intestinal tract, with PBS as a control. Images were acquired using a Nikon A1 laser-scanning confocal microscope with a 20× objective lens, dichronic mirrors (DM457/514), and two bandpass emission filters (482/35 for cyan fluorescent protein, CFP, 540/30 for yellow fluorescent protein, YFP), as described previously [26]. The YFP/CFP ratio was obtained by excitation at 458 nm. Images of purified spleen cells in PBS were obtained using the same method. Acquired images were analyzed using NIS-Elements software (Nikon, Tokyo, Japan).

### 2.5. In Vivo Stimulation Assay

To observe PPs, the peritoneal cavity of each mouse was injected with 200 μg of *Euglena* or paramylon in PBS (1 mg/mL), with PBS as a control. After 2 hours, the mice were subjected to intravital imaging analysis, as described previously [31].

### 2.6. Statistical Analysis

Statistical analysis was performed with Pearson’s chi-square test to compare the proportions of cells. R version 3.4.1 were used to conduct the statistical analyses. *p* < 0.05 was considered significant.

## 3. Results

### 3.1. Euglena and Paramylon Induce Ca^2+^ Signaling in the IECs of Mice With Ubiquitous Yc3.60 Expression

To examine whether *Euglena* and paramylon directly stimulate IECs, we carried out intravital Ca^2+^ imaging. *Euglena* induced transient Ca^2+^ signaling in most IECs (Figure 1a) and intracellular Ca^2+^ levels increased following stimulation (Figure 1b). Conversely, paramylon induced robust but sparse Ca^2+^ signaling limited to minor IEC subpopulations (Figure 1). Together, these findings suggest that *Euglena* and paramylon directly stimulate IECs.

### 3.2. Euglena and Paramylon Induce Ca^2+^ Signaling in DCs

To assess the potential immune-stimulatory effects of *Euglena* and paramylon on immune cells in vivo, we performed intravital Ca^2+^ imaging on PPs in YC3.60^flox^/CD11c-Cre mice [31] injected intraperitoneally (IP) with *Euglena* or paramylon (1 mg/mL). After 2 hours of intraperitoneal administration, intravital imaging analysis showed that *Euglena* increased intracellular Ca^2+^ levels in DCs, with 31.2% of DCs exhibiting higher intracellular Ca^2+^ concentrations (Figure 2). However, paramylon induced robust Ca^2+^ signaling in 78.2% of DCs, thus exerted stronger effects than *Euglena* (Figure 2b). These results indicate that *Euglena* and paramylon possess immune-stimulatory functions.

### 3.3. Euglena Elicits In Vitro Ca^2+^ Signaling in DRG-Derived Neurons From YC3.60^flox^/Nestin-Cre Mice

To test whether *Euglena* or paramylon have the potential to stimulate sensory neurons in the intestine, we performed Ca^2+^ imaging on primary neurons dissected from the DRG of YC3.60^flox^ /Nestin-Cre mice [32] and cultured on a dish for several days [35]. *Euglena* induced robust Ca^2+^ signaling in the DRG neurons (Figure 3a), whereas no Ca^2+^ signaling was induced by paramylon (Figure 3b), indicating that *Euglena* stimulates sensory neurons but its component paramylon does not.

## 4. Discussion

This study successfully visualized *Euglena*- and paramylon-mediated Ca^2+^ signaling in IECs, DCs, and DRG-derived neurons using Ca^2+^ imaging in YC3.60 mice. *Euglena* and paramylon both exhibited stimulatory activities in IECs in vivo and possessed immune-stimulating properties against DCs in PPs in vivo. Furthermore, *Euglena* directly induced Ca^2+^ signaling in DRG-derived neurons.

Understanding the mechanisms underlying the interaction between food compounds and IECs is important for evaluating the physiological benefits of food compounds, since stimulated IECs can produce cytokines and/or peptide hormones [30]. Probiotic microbes bind with a series of pattern recognition receptors, including toll-like receptors, nucleotide-binding sites, leucine-rich repeat-containing receptors, and retinoic acid-inducible gene-I-like receptors [37]. On IECs, the β-glucan receptor Dectin-1 triggers the secretion of pro-inflammatory cytokines, such as Interleukin-8 (IL-8) and monocyte chemoattractant protein 1 (CCL2) [22]. Although enteroendocrine cells make up less than 1% of the IEC population, they form the largest endocrine organ in the body and secrete multiple peptide hormones such as ghrelin, serotonin (5-hydroxytryptamine), Cholecystokinin (CCK), peptide tyrosine-tyrosine (PYY), glucagon-like peptide-1 (GLP-1) and glucose-dependent insulinotropic peptide (GIP) [38]. In response to food intake, enteroendocrine cells produce GLP-1, a 30-amino acid peptide hormone that exerts various metabolic actions, such as reducing appetite and food intake [39]. In addition, GLP-1 promotes insulin secretion and may contribute toward improved insulin sensitivity [40]. Thus, food components and dietary supplements that modulate nutrient-sensing pathways may have therapeutic potential for treating obesity and metabolic diseases [38,41]. A previous study showed that *Bacillus subtilis* var*. natto*, which has been shown to modulate immune responses, triggers gradual and sustained Ca^2+^ signaling in IECs [32]. Conversely, the probiotic *Lactococcus lactis,* which is similar to commensal bacteria, does not induce Ca^2+^ signaling in IECs, likely due to hyporesponsiveness following chronic exposure to bacteria [32]. In this study, we found that both *Euglena* and paramylon evoked Ca^2+^ signaling in IECs, similar to other prebiotics, suggesting that they can directly access IECs. It should be noted that *Euglena* and paramylon showed different bioactivity for IECs. *Euglena* stimulation evoked transient Ca^2+^ signaling throughout IECs with distinct Ca^2+^ signaling kinetics to those evoked by paramylon. Previous studies have shown that oral paramylon and *Euglena* intake exert preventive effects against obesity, likely due to the presence of paramylon [17,18]. Here, paramylon induced sparse transient Ca^2+^ signaling in minor IEC subpopulations, with a similar spatial pattern of Ca^2+^ signaling to the cellular distribution of enteroendocrine cells [42]. Thus, nutrient sensing in the intestinal tract may transmit neuronal signals to the brain by secreting multiple gastrointestinal hormones to modulate the physiological response to food components. However, further studies of specific cell types are required to clarify the biological properties of paramylon.

Biologically-active polysaccharides may be a potential method of preventing dysfunctional immune homeostasis [9,43]. For instance, β-glucan extracted from the maitake mushroom (*Grifola frondosa*) has been shown to control the cytokine balance between T lymphocyte Th-1 and Th-2 subsets, resulting in enhanced cellular immunity [8]. In addition, the subcutaneous application of β-1,3-glucan extracted from *Saccharomyces cerevisiae* in 20 children with asthma increased serum levels of the anti-inflammatory cytokine IL-10 and simultaneously reduced symptom scores [10]. The oral intake of paramylon has been found to reduce cytokine secretion and relieve arthritis symptoms in mice by modulating Th17 immunity [44], while studies have suggested that *Euglena* and paramylon could reduce upper respiratory tract infection symptoms and protect against influenza virus infection [13,45]. Here, we found that IP *Euglena* and paramylon injection stimulated DCs in PPs in vivo, with stronger Ca^2+^ signaling observed with paramylon treatment; thus, paramylon may be the active component of *Euglena* that affects the immunological system. This finding is consistent with previous studies indicating that β-1,3-glucans stimulate macrophages and DCs via a Dectin-1-dependent pathway [20,22]. Orally administered β-glucans are thought to access gut immune cells, such as macrophages and DCs, by being taken up into M cells, a type of IEC that lies over PPs [23,24]. On macrophages, the β-glucan receptor Dectin-1 specifically binds to paramylon and exerts immune stimulatory effects [20,46]. Alongside phagocytosis in macrophages, Dectin-1 also induces proinflammatory cytokine secretion [47]. Due to its crystalline structure, paramylon typically exists in the form of insoluble granules that are 2–3 μm in size, suggesting that paramylon may be transported across IECs via the same mechanism as pathogenic bacteria. Our findings, together with those of previous studies, suggest that oral paramylon intake could stimulate intestinal DCs in PPs to enhance host immune function. Consistently, some studies in human subjects suggest that *Euglena* and paramylon may support a healthy immune system and protect overall health [45,48]. Intravital imaging using several transgenic mice with biosensors expressed in the intestinal epithelium, nervous system, and immune cells will be a powerful tool to dissect the mechanism of a beneficial effect of the active ingredient in food products on human immunity [32].

In this study, we demonstrated that *Euglena*, but not paramylon, directly triggers Ca^2+^ signaling in DRG neurons, suggesting that *Euglena* can excite visceral afferents. Although paramylon has various bioactive functionalities [21]*,* other bioactive components of water-soluble fraction seem to activate the neurons. Indeed, the water extract partially purified from *Euglena* also contains bioactive materials, as this is crucial for preventing lung carcinoma growth and intracellular lipid accumulation [19,49]. The sensory innervation of the small intestine is due to spinal and vagus nerves, which have cell bodies in the DRG and nodose ganglion, respectively [50]. DRG afferents are largely peptidergic and express the calcitonin gene-related peptide Substance P and/or transient receptor potential vanilloid 1 [51,52]. The calcitonin gene-related peptide (CGRP) neuropeptide modifies macrophages, DCs, and other immune cells, suggesting that it plays a key role in neuro-immune cross-talk and allows sensory fibers to mediate immune function [52]. The release of neuropeptides from sensory afferents has been associated to nociceptive transmission, energy homeostasis, and longevity [53,54], whereas the vagal afferent is required for the activity of sympathetic nerves innervating brown adipose tissue triggered by a capsaicin analog, indicating that sensory afferents play important roles in inducing autonomous nerve activity [55]. Some prebiotic bacteria have been shown to alter emotional behavior by regulating the vagal nerve [56], whereas capsaicin in hot peppers can trigger Ca^2+^ signaling in the intestinal tract, which is transmitted to the nervous system and results in a transient shift toward higher arousal levels in the brain [35]. Further studies will be required to better understand the physiological role of food compound-mediated visceral afferents in homeostasis, behavior, emotion or cognitive function. Although the mechanisms underlying our observations remain unclear, the ability of *Euglena* to excite neurons may underlie its beneficial effects on health-related Quality of Life observed in human subjects [45]. In the near future, after the components involved in *Euglena* responsible for these signaling responses have been identified and a method for extracting the active ingredients has been established, *Euglena* can be utilized to produce useful ingredients on an industrial scale [57].

## 5. Conclusions

Interactions between food and the intestinal tract transmit signals via the gut–immune–brain axis by secreting multiple cytokines and hormones that modulate physiological homeostasis [38,58]. Thus, our findings help to elucidate the mechanisms via which *Euglena* and paramylon exert various effects from the intestinal tract.

## Figures and Tables

**Figure 1 nutrients-12-02293-f001:**
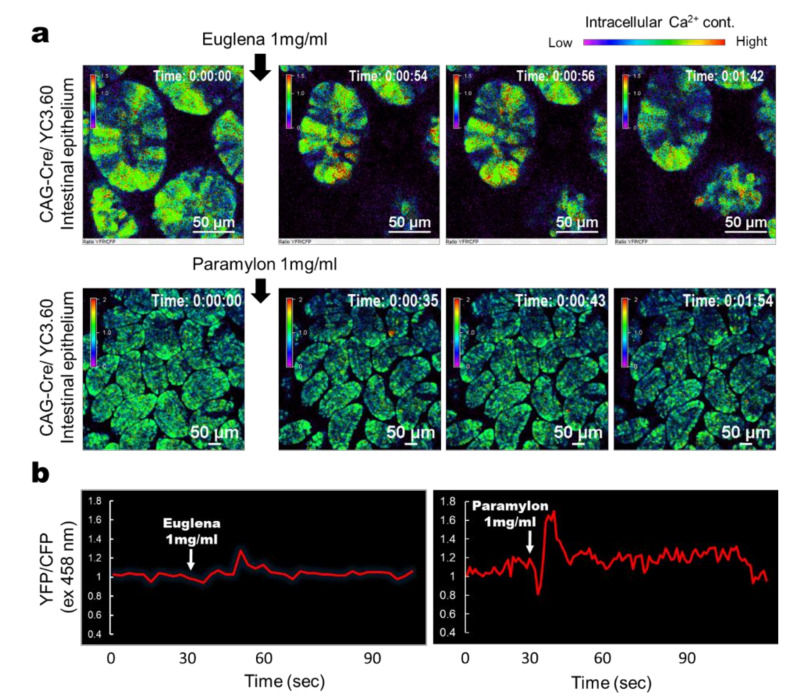
Intravital calcium (Ca^2+^) signaling mediated by *Euglena gracilis* (*Euglena*) or paramylon in the intestinal tract. (**a**) Representative ratiometric Ca^2+^ signaling images from the intestinal tract of a YC3.60^flox^/CAG-Cre mouse with ubiquitous YC3.60 expression showing yellow fluorescent protein/cyan fluorescent protein (YFP/CFP) intensity at 458 nm excitation. *Euglena* or paramylon (0.1 mL) in phosphate buffered saline (PBS) (1 mg/mL) was added at the indicated time point. The color scale indicates relative Ca^2+^ concentration. (**b**) Time course of YFP/CFP fluorescence intensity at 458 nm excitation. *Euglena* or paramylon (0.1 mL) in PBS (1 mg/mL) was added at the indicated time point. Results are representative of at least three independent experiments (*n* = 3 mice; scale bars, 50 μm).

**Figure 2 nutrients-12-02293-f002:**
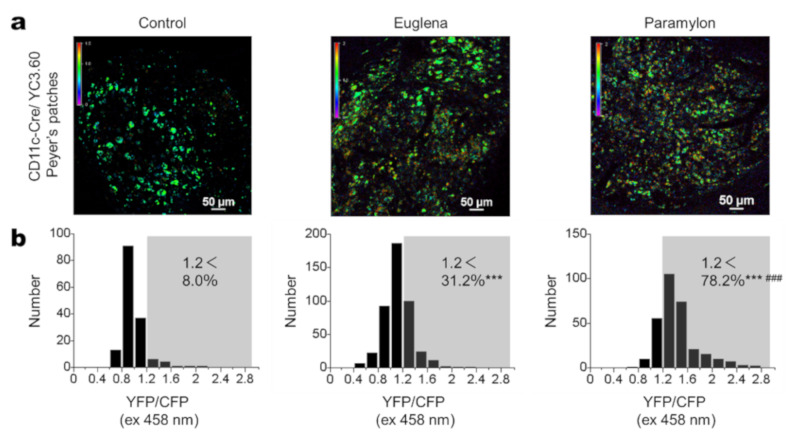
Intravital Ca^2+^ signaling in Peyer’s patches (PPs). (**a**) Representative Ca^2+^ signaling images of PPs in YC3.60^flox^/CD11c-Cre mice intraperitoneally injected with PBS (control, left) *Euglena*/PBS (center), or paramylon/PBS (right). Intravital ratiometric imaging was carried out 2 hours after injection and shows YFP/CFP excitation at 458 nm. The results are representative of at least three independent experiments (*n* = 3 mice; scale bars, 50 μm). (**b**) Distribution of intracellular Ca^2+^ levels in randomly selected cells (control, *n* = 152; *Euglena*, *n* = 455; paramylon, *n* = 303). YFP/CFP > 1.2 was defined as cells of relatively high Ca^2+^ concentration. Pearson’s chi-square test, *** *p* < 0.001 vs. Control, ^###^
*p* < 0.001 *Euglena* vs. Paramylon.

**Figure 3 nutrients-12-02293-f003:**
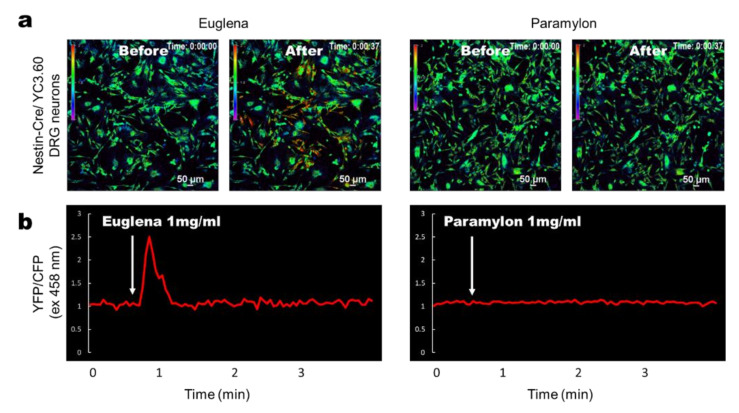
Ca^2+^ signaling images using *Euglena* and Paramylon in dorsal root ganglia (DRG) neurons in vitro. (**a**) Representative ratiometric Ca^2+^ signaling images in DRG cells from YC3.60^flox^/Nestin-Cre mice showing YFP/CFP excitation at 458 nm. *Euglena* or paramylon (0.1 mL) in PBS (1 mg/mL) was added to the cell culture at the indicated time point. The color scale indicates relative Ca^2+^ concentration. (**b**) Time course of ratiometric YFP/CFP fluorescence intensity at 458 nm excitation. Results are representative of at least three independent experiments (*n* = 9 mice; scale bars, 50 μm).

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
