# Peer review of "Euglena Gracilis and β-Glucan Paramylon Induce Ca2+ Signaling in Intestinal Tract Epithelial, Immune, and Neural Cells"

_nutrients, 2020, doi:10.3390/nu12082293_

Round 1

Reviewer 1 Report

The aim is stated clear. The authors stated clearly what study found and how they did it.

The title is informative and relevant. The references are relevant and recent. Appropriate and key studies are included. The introduction reveals what is already known about this topic. The research question is clearly outlined. 

There are multiple study methods, valid and reliable. There are enough details provided in order to replicate the study.

The data is presented in an appropriate way. Tables and figures are relevant and clearly presented. The text in the results adds to the data and it is not repetitive. Statistically significant results are clear. Results are discussed from different angles and placed into context without being overinterpreted.

The conclusions answer the aim of the study. The conclusions are supported by references and own results.

The study design is appropriate to answer the aim. The article is consistent within itself.

The major flaws of this article are these are only in vivo results that should be further investigated in clinical settings.

Overall strengths of the article are the rich methodology and the significant novel results regarding the direct effect of β-1,3-glucan on DCs in vivo and will help elucidate the mechanisms via which Euglena and paramylon exert various effects in the intestinal tract.

Specific comments on the weaknesses of the article and what could be improved:

Major points - none

Minor points 

  1. Could you please add some paragraphs in the discussion to make clear which results are with practical meaning and how they could be implemented for humans
  2. The limitations of the study are not stated

Reviewer 2 Report

The authors studied the Ca2+ signaling induced by Euglena and paramylon in intestinal epithelial cells (IECs), dendritic cells (DCs) in intestinal Peyer's patches and nerve cells in the intestine. They performed cell type-specific intravital Ca2+ imaging in vivo and in vitro among transgenic mouse lines with YC3.60 expression.

This study concludes that Euglena and paramylon, like other prebiotics, both exhibit stimulatory activities in IECs in vivo and possess immune-stimulating properties against DCs in PPs in vivo, while Euglena directly induced Ca2+ signaling in DRG-derived neurons.

The overall level of the paper is good and written nicely. The Background section provide useful information for the readers. Discussion can be improved by explaining the differences observed in results due to Euglena and Paramylon stimulation in different transgenic mice. However, some concerns are highlighted below.

Comments:

  1. How long does the effect of Euglena and Paramylon stimulation last in the intestine?
  2. Was the difference in Ca2+ concentration statistically significant in stimulated and control mice?
  3. In Fig 1a, intracellular Ca2+ levels seem to be reduced after 1:42 minutes versus 56 sec after stimulation with Euglena.
  4. Why does the stimulation with Euglena and paramylon differ for Ca2+ concentration in Fig 1? Please discuss this observation.
  5. In Fig. 2b, the graph that represents the distribution of intracellular Ca2+ levels in randomly selected cells in Euglena is not correct. The number for YFP/CFP at 1.2 is not less than 31.2%. The bar seems to be near 100%.
  6. What is the reason for the difference in induced Ca2+ signaling in DRG-derived neurons by Euglena versus Paramylon?
